# Functional group tolerant hydrogen borrowing C-alkylation

Elliot P. Bailey[1], Timothy J. Donohoe ®[1] ✉ & Martin D. Smith ®[1] ✉

Hydrogen borrowing is an attractive and sustainable strategy for carbon–carbon bond formation that enables alcohols to be used as alkylating reagents in place of alkyl halides. However, despite intensive efforts, limited functional group tolerance is observed in this methodology, which we hypothesize is due to the high temperatures and harsh basic conditions often employed. Here we demonstrate that room temperature and functional group tolerant hydrogen borrowing can be achieved with a simple iridium catalyst in the presence of substoichiometric base without an excess of reagents. Achieving high yields necessitates the application of anaerobic conditions to counteract the oxygen sensitivity of the catalytic iridium hydride intermediate, which otherwise leads to catalyst degradation. Substrates containing heteroatoms capable of complexing the catalyst exhibit limited room temperature reactivity, but the application of moderately higher temperatures enables extension to a broad range of medicinally relevant nitrogen rich heterocycles. These newly developed conditions allow alcohols possessing functional groups that were previously incompatible with hydrogen borrowing reactions to be employed.

Hydrogen borrowing methodology employs alcohols as alkylating agents via a sequence of oxidation, condensation, and reduction reactions, using a catalytic metal to mediate the process[1–3]. In principle, this is an attractive strategy in comparison to the traditional alkylation of stoichiometrically formed enolates using alkyl halides: alcohol substrates are commercially abundant and stable, the process is catalytic in a weaker base, and water is generated as the sole by-product (Fig. 1a). This process is exemplified by the alkylation of acetophenone with benzyl alcohol, which has been achieved with a wide range of different metal catalysts[4,5]; the α-alkylation of ketones via hydrogen borrowing has been intensively investigated[6–10]. However, most hydrogen borrowing[11] C–C bond forming reactions operate under a high temperature regime (80–180 °C) in the presence of stoichiometric or superstoichiometric base, and require extended reaction times and an excess of alcohol to enable high conversions[12,13]. These conditions are reflected in the low catalyst productivity[14] and limited functional group tolerance of most hydrogen borrowing reactions that form carbon–carbon bonds: complex molecules, sensitive or reactive functional groups and nitrogen-containing heterocycles are generally not reported to be effective substrates. There are few examples of

carbon-carbon bond-forming hydrogen borrowing reactions that operate at ambient temperature[15]. Quintard, Rodriguez and co-workers reported the room temperature enantioselective alkylation of ketoesters with allylic alcohols and diketones with an earth abundant catalyst, demonstrating success with alkyl chlorides and some heterocycles in a functional group tolerance study[16–19]. Glorius and co-workers reported an α-alkylation of ketones with alcohols at 40 °C, including alkyl bromide and chloride functional groups at extended reaction times (2 days)[20]. Zhao and co-workers reported an elegant room temperature Guerbet reaction (Fig. 1b) that employed alkyl alcohols which notably demonstrated pyridines and terminal alkenes as viable functionality[21]. We considered whether there was any intrinsic reason why hydrogen borrowing should require such elevated temperatures and harsh conditions. A generalised catalytic cycle for metal catalysed hydrogen borrowing alkylation of a ketone (Fig. 1c) illustrates that the process broadly consists of ligand exchange with an alcohol to generate a metal alkoxide that undergoes oxidation via β-hydride elimination to liberate an aldehyde and a metal hydride species. This aldehyde is trapped by a base-catalysed aldol-elimination sequence to produce an enone that is then reduced by the in-situ

[1]Chemistry Research Laboratory, University of Oxford, Oxford, UK. ✉e-mail: timothy.donohoe@chem.ox.ac.uk; martin.smith@chem.ox.ac.uk

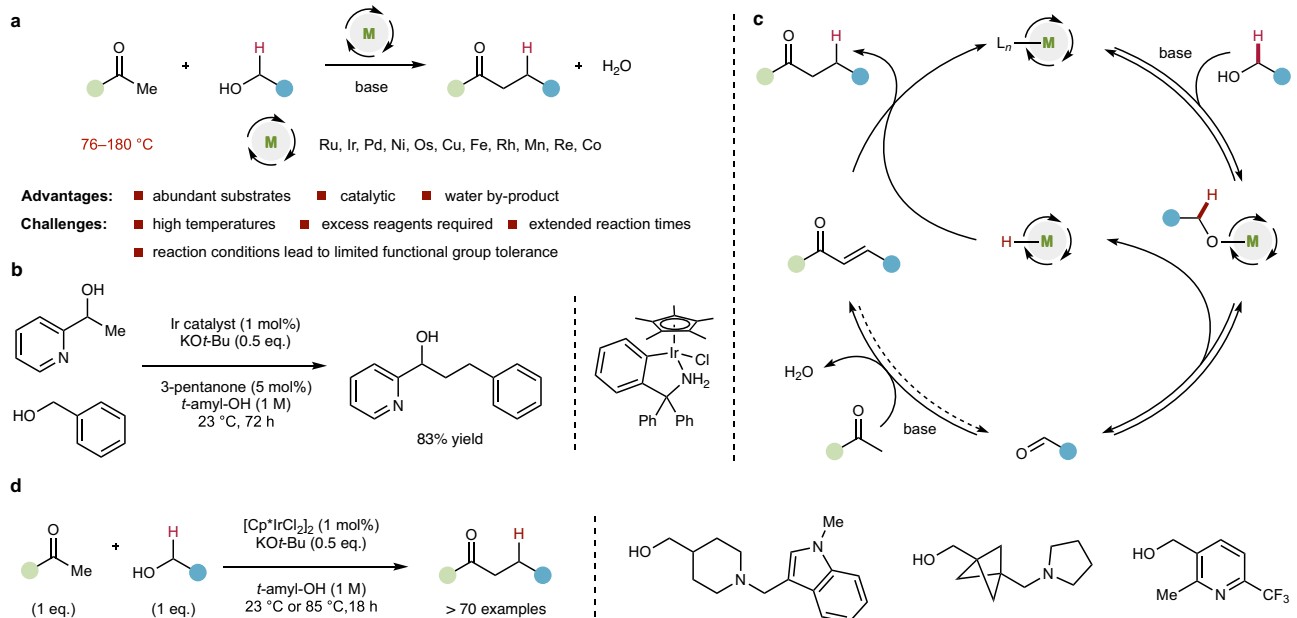

**Fig. 1 | Hydrogen borrowing alkylation generally has limited functional group tolerance. a** Hydrogen borrowing alkylation with alcohols has significant advantages over traditional alkylation of stoichiometric enolates with alkyl halides but is also limited by high temperatures and harsh reaction conditions that lead to limited functional group tolerance. **b** State-of-the-art in room temperature C-C bond forming hydrogen borrowing Guerbet reaction from Zhao et al. (*Angew. Chem. Int. Ed.* **59**, 11384–11389 (2020)). **c** Generalized C-C bond forming hydrogen borrowing catalytic cycle, in which a key metal hydride intermediate is generated during the reaction. **d** This work: functional group tolerant hydrogen borrowing alkylation. M = metal; Cp* = pentamethylcyclopentadienyl.

generated metal hydride leading to the alkylation product, while regenerating the catalytic metal species. The necessity for one carbonyl component in this reaction manifold to be non-enolizable to enable selective cross-aldol condensation is recognized[20], but there are no clear mechanistic reasons why this reaction cannot be applicable to a broad range of substrates. We hypothesized that lower temperatures should enable greater functional group tolerance. Within this general mechanistic regime, we reasoned that catalyst complexation/sequestration by metal-binding heteroatom-containing substrates may kinetically inhibit the oxidation step by preventing effective β-hydride elimination. We also recognised that metal hydrides have been demonstrated to possess sensitivity to oxygen[22-27], leading to metal hydroperoxide and higher oxidation state oxo-species, which could lead to catalyst degradation or inhibit the hydrogen borrowing process; this has been exploited in interrupted hydrogen borrowing reactions[28-30]. In this study we show that functional group tolerant hydrogen borrowing alkylation can be achieved by tuning reaction conditions using a commercially available iridium catalyst in the presence of substoichiometric base under anaerobic conditions. We also demonstrate that the presence of oxygen in the reaction mixture leads to the generation of a catalyst that is unreactive at room temperature but is notably still effective in hydrogen borrowing reactions at higher temperatures (85 °C and 115 °C).

## Results

Our initial focus was on exploring and developing the functional group tolerance of a range of different alcohols and we commenced our study using pentamethylphenyl (Ph*) methylketone **1** and heptanol **2** as substrates. Ph* methylketone **1** is a privileged substrate for hydrogen borrowing reactions since unwanted electrophilic reactions (including reduction) of the carbonyl are discouraged due to steric shielding by the *ortho*-methyl groups[31]. We explored a range of different conditions and catalysts (see supplementary information p7 for full discovery and optimization) in our search for a process that gave high conversion to monoalkylated product at room temperature. We observed excellent reactivity with 1 mol%

[Cp*IrCl₂]₂ in the presence of half an equivalent of KOt-Bu; this enabled a 1:1 ratio of ketone and alcohol starting materials to be employed (90% isolated yield). Crucially we noted that anaerobic conditions were necessary for proficient reactivity (3% yield in air). We observed a similar result with 2 mol% ruthenium-MACHO catalyst[32] under these conditions: 81% yield at 23 °C under anaerobic conditions and 0% yield in air. These results are consistent with the reported sensitivity of metal hydrides to oxygen and may aid in rationalizing why an excess of hydride donor (alcohol) is unnecessary in our optimized system.

### Enabling new functional group tolerance in hydrogen borrowing alkylation at room temperature

With this result in hand, we decided to examine the room temperature functional group tolerance with [Cp*IrCl₂]₂ as catalyst under the conditions described above (Fig. 2). Heptanol **2** (90% yield) and benzyl alcohol **3** (96% yield) were both highly effective substrates, leading to alkylation products in high yields. β-Substitution of the alcohol was well tolerated, as exemplified by high conversion of 2-methylbutan-1-ol **4** and 2-cyclohexylpropan-1-ol **5** to products (88% and 81% yield respectively). Benzylic ethers such as **6** are well tolerated at room temperature (90% yield), as were TBS silyl ethers **7** (94% yield). We also conducted some reactions at 85°C to assess whether higher reaction temperatures compromised reactivity or yield. TBS silyl ether substrate **7** at 85 °C delivered the alkylated product in diminished yield (26% yield), likely due to desilylation to generate a diol which has demonstrated incompatibility in this reaction (see also **31**–**34**). Cyclopropylmethanol **8** and (2,2-difluorocyclopropyl)methanol **9** both performed well at room temperature (94% and 59% yield respectively) whilst both gave significantly poorer yields at higher temperatures (46% and 14% yield respectively), consistent with the enabling of undesirable side reactions (see supplementary information p16). Sensitive functionality such as oxetanes **10**, alkyl bromides **11** and alkyl chlorides **12** were also well tolerated in the room temperature process (94%, 65% and 89% yield respectively) whilst diminished yields were observed at higher temperature consistent with elimination or other

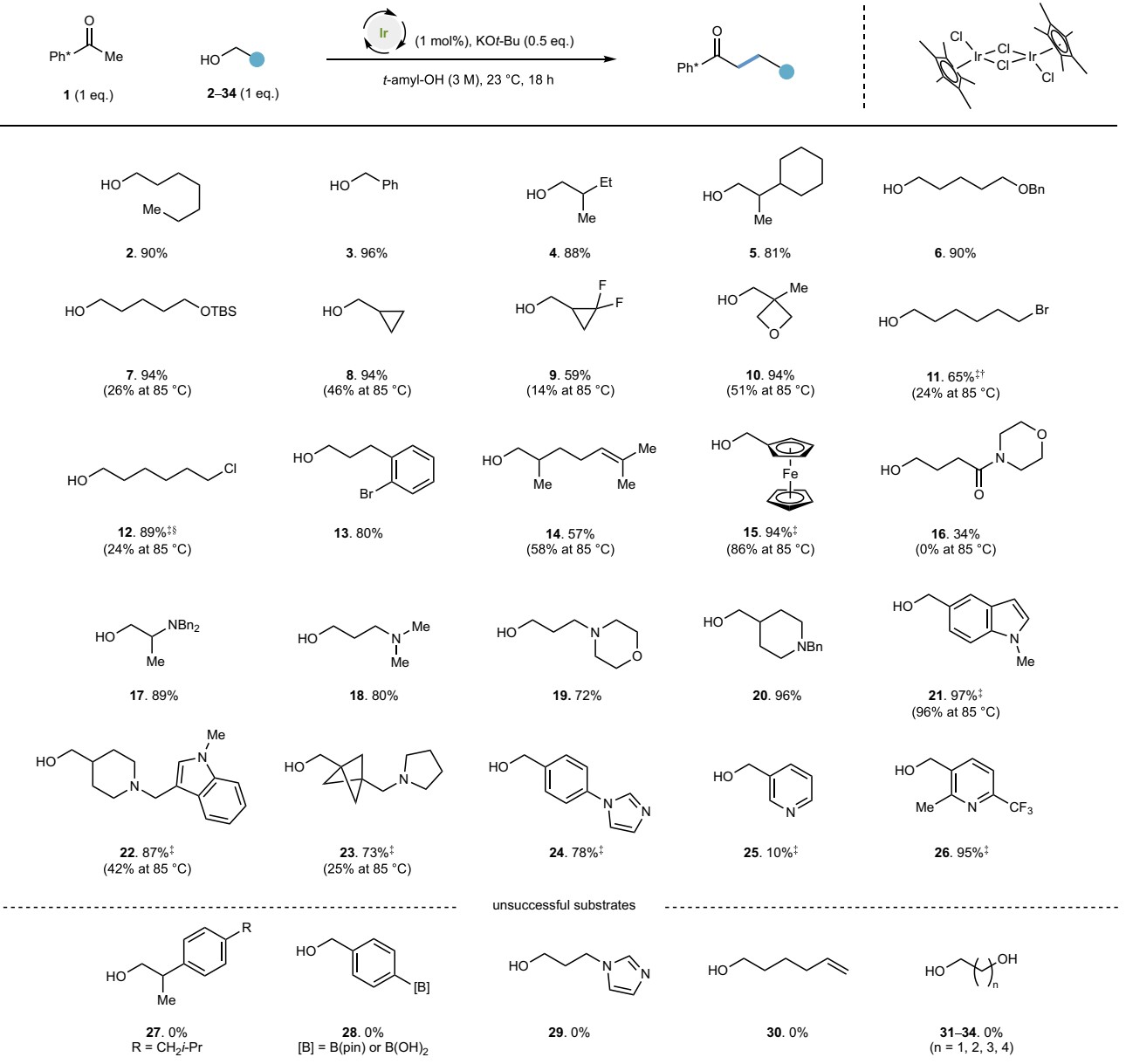

**Fig. 2 | Enabling new functional group tolerance in hydrogen borrowing alkylation at room temperature.** Conditions: Ph* methyl ketone (0.3 mmol), alcohol (0.3 mmol), [Cp*IrCl₂] (1 mol%), KOt-Bu (0.15 mmol), t-amyl-OH [substrate] = 3 M, degassed (N₂ atmosphere), 23 °C, 18 h. Yields reflect isolated products except for yields at 85 °C that were determined by quantitative ¹H NMR spectroscopy. ‡Reaction performed at [substrate] = 1 M. †Reaction performed with 2 mol% catalyst. §Yield determined by quantitative ¹H NMR spectroscopy. TBS *tert*-butyldimethylsilyl. Ph* pentamethylphenyl; pin pinacol.

side-reactions (see incompatibility of mono-substituted alkenes such as **30**). Aryl bromides such as **13** were tolerated in the reaction (80% yield) and a trisubstituted alkene **14** also performed well, without any evidence of hydrogenation or alkene isomerisation.

Ferrocenylmethanol **15** proved a good substrate in this reaction, both at room temperature and higher temperature (94% and 86% yield respectively) and an amide-containing alcohol **16** afforded products in moderate yield (34%) whilst this substrate was ineffective at higher temperature. Nitrogen substitution at the β- and γ- positions to the alcohol was well tolerated, exemplified by β-dibenzylamino **17** and γ-dimethylamino **18** substrates (89% and 80% yields respectively). Morpholino-alcohol **19** and piperidine **20** both performed excellently (72% and 96% yield). Indoles were well tolerated, as demonstrated by benzylic and alkyl alcohols (**21** 97% and **22**, 87% respectively). A neopentylic alcohol with a bicyclo[1.1.1]pentane (BCP) group **23**, an architecture which has shown increasing utility as a 1,4-phenyl bioisostere

performed well at room temperature (73%) but poorly at higher temperature (25% yield). High product yields were also observed with an alcohol substrate containing an aryl imidazole **24** (78% yield).

A simple pyridyl substrate **25** gave only a low yield of product (10%) while electron-deficient pyridine **26** was almost quantitatively converted to product (95% yield). We were also able to outline some limitations of this method: ibuprofen alcohol **27** demonstrated the intolerance of β-phenyl substitution in the alcohol (0% yield), likely a consequence of formation of a stable enolate after oxidation to an aldehyde. Boronic acids and boronic esters **28** were not tolerated at room temperature or at 85 °C. Imidazolyl alcohol **29** was unsuccessful, likely a consequence of E1cB elimination of imidazole from the corresponding aldehyde intermediate. An alcohol containing a monosubstituted alkene such as **30** was not tolerated in the reaction either at room temperature or at higher temperatures. Diols **31**–**34** are not successful substrates; alcohol consumption was observed but

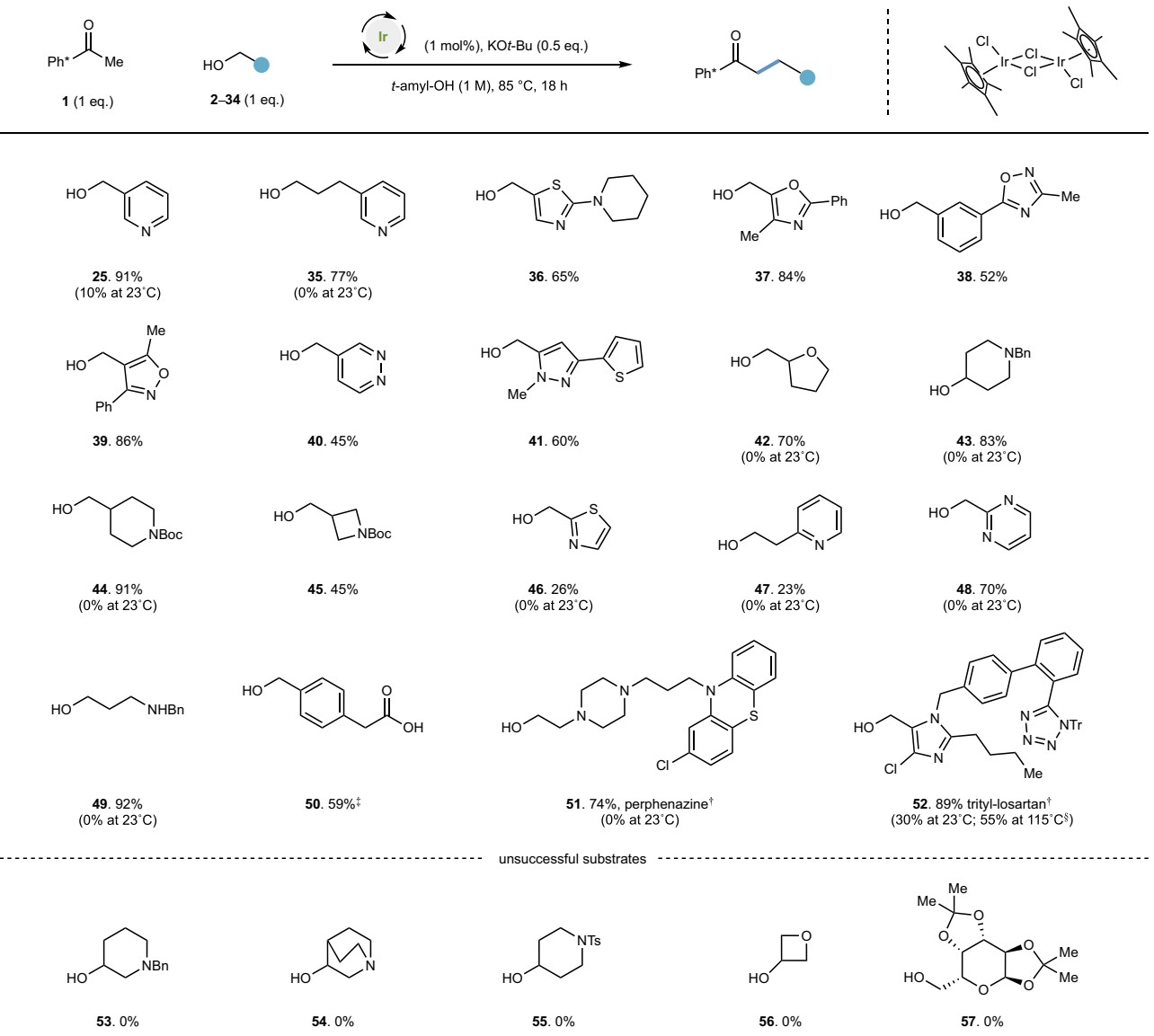

**Fig. 3 | Coordinating and heterocyclic alcohols require higher temperature to be successful alkylating agents.** Conditions: Ph* methyl ketone (0.3 mmol), alcohol (0.3 mmol), [Cp*IrCl₂] (1 mol%), KOt-Bu (0.15 mmol), t-amyl-OH [substrate] = 1 M, degassed (N₂ atmosphere), 85 °C. Yields reflect isolated products. ‡Product isolated as methyl ester; 1.5 eq. KOt-Bu employed. §Yield determined by quantitative ¹H NMR spectroscopy. † Reaction performed on 0.1 mmol scale. Tr trityl, Bn benzyl, Ts p-toluenesulfonyl.

neither monoalkylated or cyclised products were observed in significant yields.

That the iridium catalyst can be inhibited by coordination is consistent with the room temperature conversion of pyridines **25** and **26**. Whilst relatively electron rich pyridine **25** is a poor room temperature substrate (10% yield), decreasing the Lewis basicity of the pyridyl nitrogen (both sterically and electronically) as in substrate **26** enables a proficient reaction at room temperature (95% yield). We postulated that highly coordinating substrates may require higher temperatures to enable the reaction to proceed[33]. Thus, treatment of the pyridine **25** under the same anaerobic conditions developed previously but now at 85 °C, yielded the hydrogen borrowing alkylation product in 91% yield (Fig. 3).

## Coordinating and heterocyclic alcohols require higher temperature to be successful alkylating agents

With this result in hand, we now examined whether this would enable an even wider range of heterocyclic and functionalized alcohols to be employed in the reaction (Fig. 3).

A non-benzylic pyridyl alcohol **35** proved to be an excellent substrate under these conditions (77%). Thiazole **36** and oxazole functionality **37** were well tolerated in the reaction (65% and 84% yields respectively). 1,2,4-Oxadiazole **38** and isoxazole groups **39** were also effective substrates (52% and 86% yields, respectively); no evidence of N-O cleavage was observed. Similarly, pyridazine **40** and pyrazole-containing alcohols **41** were viable substrates (45% and 60% yields respectively) and N-N bond cleavage products were not observed. Tetrahydrofuranyl alcohol **42** gave poor room temperature conversion but at this higher temperature yielded product in 70% yield. Secondary alcohols such as **43** also delivered alkylated products in excellent yield (83% yield); this substrate was ineffective at room temperature, which may reflect challenges in the aldol condensation. A Boc-protected piperidine alcohol **44** gave the product in excellent yield (91%), and Boc-protected azetidine alcohol **45** was a moderately successful substrate (45%). Alcohols bearing groups that are capable of bidentate coordination to iridium performed more poorly in the reaction: thiazol-2-ylmethanol **46** and 2-(pyridine-2-yl)ethan-1-ol **47** both yielded moderate amounts of product (26% and 23% yield respectively).

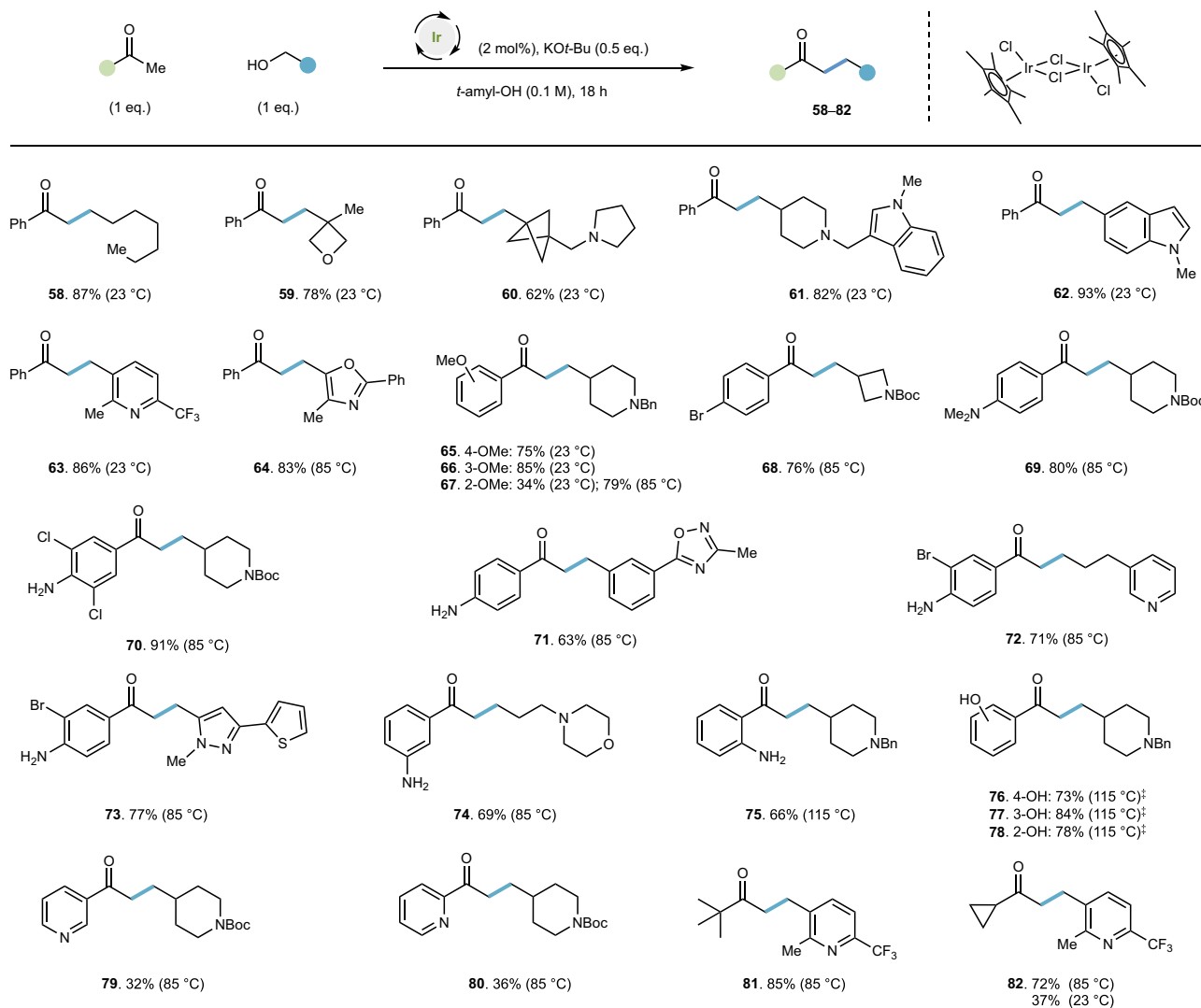

**Fig. 4 | Broadening substrate scope to encompass functionalized ketones.** Conditions: ketone (0.3 mmol), alcohol (0.3 mmol), [Cp*IrCl$_2$] (2 mol%), KO$t$-Bu (0.15 mmol), $t$-amyl-OH [substrate] = 0.1 M, degassed (N$_2$ atmosphere), temperature as indicated. Yields reflect isolated products. ‡1.5 eq. KO$t$-Bu employed. Boc $tert$-butoxycarbonyl, Bn benzyl.

However, pyrimidin-2-ylmethanol **48** proved effective in the reaction (70% yield), whilst displaying no reactivity at room temperature. Amino alcohol **49** was completely ineffective at room temperature but afforded product in 92% yield at higher temperature; carboxy-substituted benzyl alcohol **50** was also successful (59% yield). Per-phanazine **51**, an antipsychotic drug with an electron rich nitrogen β-to the alcohol was successfully alkylated at 85 °C (in 74% yield); this substrate again proved ineffective at room temperature. *N*-Trityl losartan **52**, a derivative of a World Health Organization essential medicine for hypertension, proved effective at 85 °C (89%). For this substrate diminished yields were observed at both room temperature and higher temperatures (23 °C: 30%; 115 °C: 55% yields). This is consistent with the idea that (i) high temperatures can lead to non-productive degradation of sensitive functionality; and (ii) coordinating substrates require thermal activation to be rendered effective. At this higher temperature, we observed that a range of β-heteroaromatic secondary alcohols **53**–**56** and a D-galactopyranose derivative **57** were unsuccessful substrates.

### Broadening substrate scope to encompass diverse ketones
Having established that a wide range of different functionalized alcohols can be viable alkylating agents at various temperatures when used in conjunction with the privileged Ph* methyl ketone, we investigated whether the combination of other ketones and a range of these investigated alcohols were possible (Fig. 4). We found that acet-ophenone was an excellent room temperature coupling partner with heptanol (yielding **58** in 87% yield) when the reaction was run anae-robically at a lower concentration (Fig. 4, see supplementary infor-mation p46 for optimization). These conditions are broadly applicable to a range of functionalized alcohols. Oxetane-containing ketone **59** can be generated in good yield (78%) and functionalized BCP **60** is also efficiently synthesized using this method (62% yield). Indole substrates are well tolerated, leading to products **61** and **62** in 82% and 93% yield respectively. Heterocycles commonly employed in medicinal chem-istry programmes—such as pyridines (**63**, 86% yield) and oxazoles (**64**, 83% yield at 85 °C) are also viable substrates. Substitution on the aryl group of the ketone can be accommodated: 4- and 3-methoxy acet-ophenone can be coupled effectively at room temperature with amino alcohol **20** to yield **65** (75% yield) and **66** (85% yield), respectively.

2-Methoxy acetophenone gives a moderate yield of **67** at room temperature (34% yield); this is improved to 79% yield at 85 °C. We subsequently explored a range of differently substituted aryl ketones, and their reactions with heterocycle-containing alcohols. 4-Bromoacetophenone is an effective substrate, yielding azetidine-

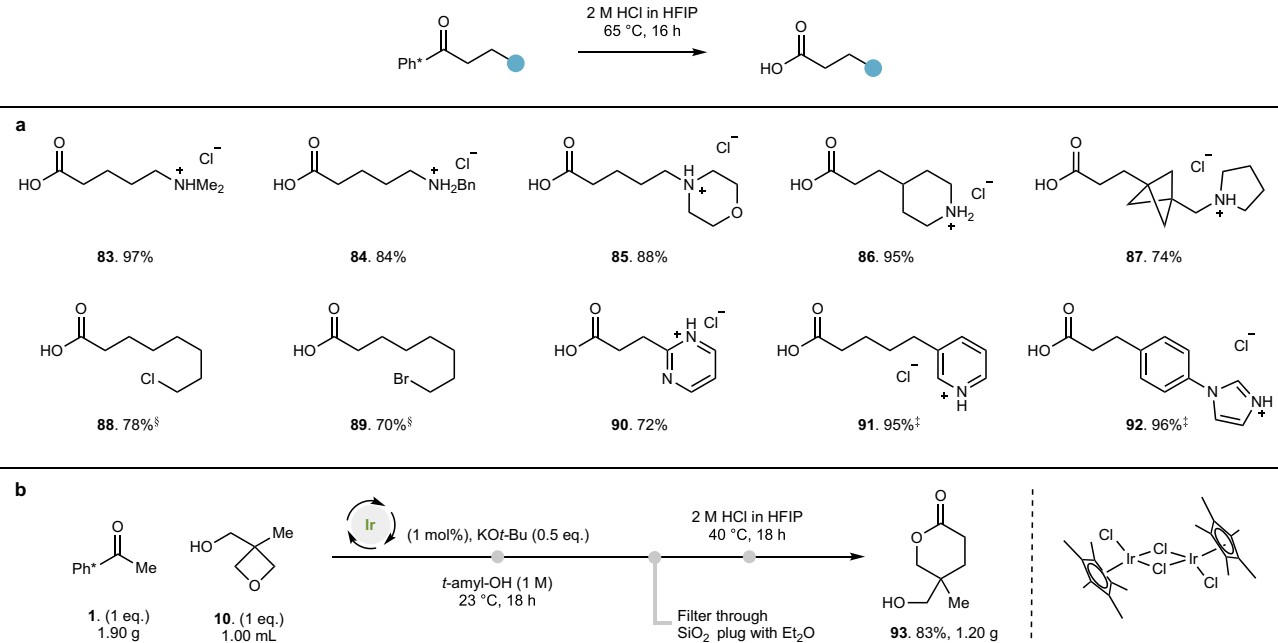

**Fig. 5 | Generation of functionalized carboxylic acids through Ph\* derivatization. a** Conditions: 2 M HCl (aq.) in HFIP (1,1,1,3,3,3-hexafluroroisopropanol), 65 °C, 16 h. **b** Scale-up and derivatization. Conditions: Ph\* methyl ketone (10 mmol), oxetane alcohol (10 mmol), [Cp\*IrCl₂] (1 mol%), KO*t*-Bu (5 mmol), *t*-amyl-OH [substrate] = 1 M, degassed (N₂ atmosphere), 23 °C, 18 h; then 2 M HCl (aq.) in HFIP, 40 °C, 18 h. ‡4 M HCl in HFIP was used. § Reaction conducted at 40 °C.

containing compound **68** (76% yield at 85 °C). We also discovered that 4-dimethylaminoacetophenone and a range of aniline-containing derivatives were well tolerated affording **69** in 80% yield and **70** in 91% yield, respectively. This can be extended to the synthesis of compounds incorporating heterocycles that are not common substrates in hydrogen borrowing methodology: 4-aminoacetophenones can be coupled with alcohols containing an oxadiazole (to afford **71**, 63% yield), a pyridine (to afford **72**, 71% yield) and a thiophene-pyrazole biaryl (to give **73**, 77% yield). 3-Aminoacetophenone and 2-aminoacetophenone are also effective enolate precursors, yielding **74** (69% yield) and **75** (66% yield at 115 °C). Pleasingly hydroxy-substituted acetophenones can also be employed in a reaction with a piperidine-containing alcohol to yield **76**–**78** in good yields; this requires higher temperatures as with **75** (115 °C) and also more base (1.5 eq.), presumably due to stoichiometric deprotonation of the phenol functional group. 3-Acetyl pyridine and 2-acetyl pyridine are also viable substrates, yielding **79** and **80** in moderate yields at 85 °C (32% and 36% respectively). Pinacolone can be efficiently coupled with pyridyl alcohol **26** to afford **81** in 85% yield; the same alcohol can be coupled with cyclopropylmethylketone to afford **82** in moderate yield at 23 °C (37%), but in higher yield (72%) at 85 °C. Ketone substrates that were ineffective at 23 °C or 85 °C generally possessed electron-deficient aryl groups (e.g., CF₃, F or CN substitution of acetophenone) or higher order cycloalkyl substitution (cyclobutyl, cyclopentyl or cyclohexyl methyl ketones; see supplementary information p60).

### Derivatization of Ph\* functional group for generation of carboxylic acids

With a series of hydrogen borrowing alkylation products in hand, we were able to demonstrate that the Ph\* group could be cleaved in a number of substrates to yield carboxylic acid derivatives through treatment with aqueous HCl in HFIP (1,1,1,3,3,3-hexafluroroisopropanol) at 40 °C or 65 °C[34]. This process is tolerant of tertiary amines, to yield **83** (97% yield) and **84** (84% yield) and heterocycles such as morpholine **85** (88% yield) and piperidine **86** (95% yield inclusive of piperidine *N*-Boc deprotection). BCP derivatives such

as **87** are tolerated under these conditions (74% yield), as are potentially reactive functional groups such as alkyl chlorides **88** (78% yield) and bromides **89** (70% yield). Substrates containing aromatic heterocycles such as pyrimidines (**90**, 72% yield), pyridines (**91**, 95% yield) and imidazoles (**92**, 96% yield) are all smoothly cleaved under these conditions. Further, substrates containing basic nitrogen functionality were isolated without column chromatography as their corresponding HCl salts. We were also able to demonstrate that these optimized conditions offer a scalable route to relatively complex materials (Fig. 5b). The reaction between ketone **1** and oxetane-containing alcohol **10** could be performed at 10 mmol scale with unaltered reaction conditions at room temperature without issue; we also demonstrated that this could be telescoped into an acidic derivatization of the Ph\* group with HCl in HFIP at 40 °C to ultimately yield lactone **93** in 83% overall yield over two steps[35].

### Preliminary mechanistic insight

Having demonstrated that a wide range of heteroatom-containing functional groups are tolerated by the hydrogen borrowing process, we further investigated the effect of oxygen on the model reaction between Ph\* ketone **1** and heptanol **2**, both at room temperature and at higher temperatures (Fig. 6a). Under anaerobic conditions (at 23 °C, 85 °C and 115 °C) high yields of the product **94** and near complete conversion of both starting materials was observed (91%, 88% and 87% yields respectively). Running the reaction under air led to a notably diminished yield at 23 °C (3% yield) and a slight recovery in performance at 85 °C (21% yield) which improves to 63% yield at 115 °C. Performing this model reaction under an oxygen atmosphere also led to similarly low conversions (see supplementary information p79). With this data in hand, we set out to (i) confirm that an iridium hydride species is formed under our optimized reaction conditions; and (ii) explore whether our assertion that that the lack of product formation at room temperature under aerobic conditions can be attributed to the reaction of this intermediate with molecular oxygen.

Under the optimized reaction conditions at 23 °C (with 100 equivalents of heptanol relative to 1 equivalent of catalyst) [Cp\*IrCl₂]₂

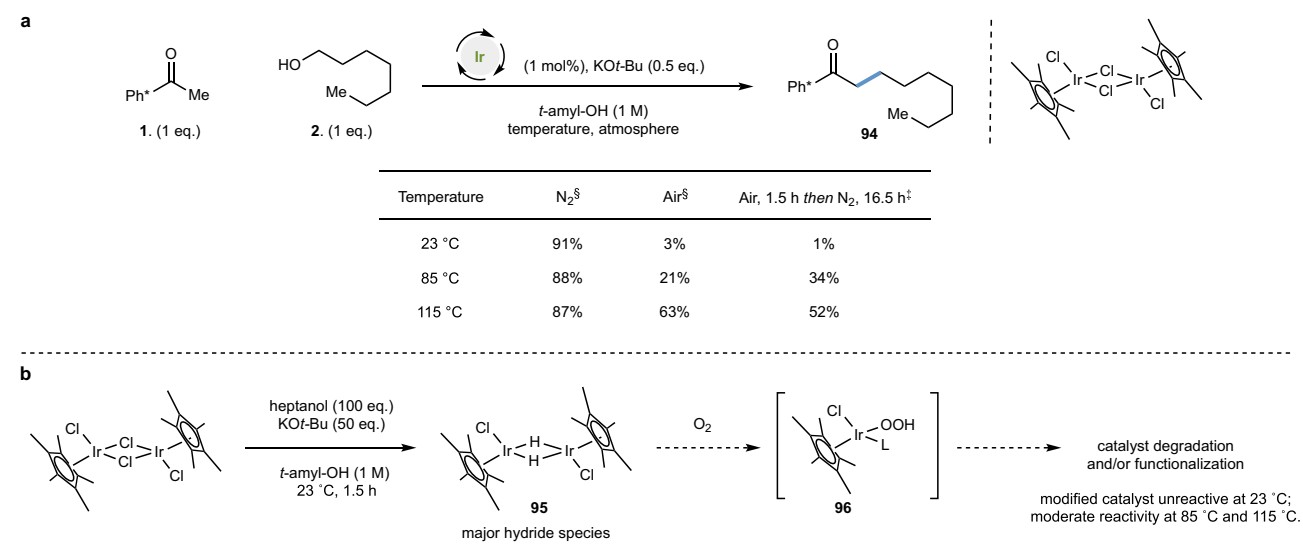

**Fig. 6 | Mechanistic studies are consistent with the presence of an iridium hydride during the reaction and its sensitivity to oxygen; the modified catalyst is unreactive at 23 °C, but can catalyse the reaction moderately at 85 °C and 115 °C. a** Reaction outcome under different atmospheres at 23°C, 85°C and 115°C. Conditions: Ph* methyl ketone (0.3 mmol), heptanol (0.3 mmol), [Cp*IrCl$_2$] (1 mol%), KO$t$-Bu (0.15 mmol), $t$-amyl-OH [substrate] = 1 M. Reactions are degassed before being stirred under the relevant atmosphere. Data in the table refers to yield of **94** as determined by quantitative $^1$H NMR spectroscopy. **b** Observation of dihydride under the optimized reaction conditions at room temperature. Conditions: heptanol (0.6 mmol), [Cp*IrCl$_2$] (1 mol %), KO$t$-Bu (0.3 mmol), $t$-amyl-OH [heptanol] = 1 M. §Reaction was degassed and stirred under the relevant atmosphere and temperature for 18 h. ‡ Reaction was stirred under air at 23 °C for 1.5 h before being degassed and stirred under nitrogen at the relevant temperature for 16.5 h (total reaction time 18 h). L = generic neutral ligand; Ph* = pentamethylphenyl.

was converted to dihydride **95** as the major hydride species (100:5:1 ratio) as indicated by $^1$H NMR spectroscopy ($\delta_H = -13.63$ ppm; Fig. 5b)[36]. Further experiments demonstrated a concentration dependency with respect to alcohol and base in determining whether a monohydride or dihydride was generated (see supplementary information p74). This is suggestive that the dihydride is the predominant species for the duration of the reaction, since alcohol equivalents decrease from 100 to 0 as the reaction progresses. This aligns with work by Ison and co-workers,[37] who observed a similar concentration dependency in their work on oxidation reactions with [Cp*IrCl$_2$]$_2$[38]. This also aligns with work by Nguyen and co-workers who outline the importance of one chloride ligand being bound to iridium in the active catalytic species in their work on transfer hydrogenation with an immobilized variant of [Cp*IrCl$_2$]$_2$[39]. Exposing solutions of dihydride **95** (or corresponding monohydride) to molecular oxygen led to complete consumption of all iridium hydride species (in <2 min), consistent with the observed lack of product formation under aerobic conditions in the room temperature catalytic reaction. The reaction of metal hydrides in situ with oxygen also has the net effect of consuming the alcohol substrate and is consistent with other studies under aerobic conditions that require an excess of this reagent[31]. We were unable to identify the products of the reaction of iridium hydrides with oxygen, consistent with observations by Ison and co-workers[28]. It has been proposed that the reaction of Cp*[Ir$^{III}$] hydrides with oxygen can generate an iridium hydroperoxide complex such as **96** that can lead to functionalization/degradation of the Cp* ligand[40–44], irreversibly changing the nature of the catalytic species. To probe this tenet, reactions between Ph* methyl ketone **1** and heptanol **2** in the presence of [Cp*IrCl$_2$]$_2$ were stirred in air for 1.5 h at 23 °C, before being transferred to anaerobic conditions and stirred at 23 °C, 85 °C and 115 °C for 16.5 h. The reaction maintained at 23 °C was inefficient (1% yield) consistent with the generation of a catalyst species that is not capable of mediating hydrogen borrowing at room temperature (under these conditions). However, at 85°C, product was generated in moderate yield (34%) which improved substantially (to 52% yield) at

115 °C. This may offer a suggestion as to why previously disclosed hydrogen borrowing enolate alkylation studies employing [Cp*IrCl$_2$]$_2$ require high temperatures to attain good reactivity with relatively benign substrates.

We have demonstrated that hydrogen borrowing C-alkylation reactions can be remarkably functional group tolerant using only one equivalent of alcohol at a range of temperatures. Developing a room temperature reaction allowed the inclusion of functional groups which are otherwise sensitive to base-mediated side reactions at higher temperatures. We also demonstrated that potentially coordinating functionality such as medicinally relevant nitrogen-rich heterocycles necessitate higher reaction temperatures to become proficient substrates. Maintaining anaerobic reaction conditions enables room temperature reactivity by preventing reaction of a key iridium hydride intermediate with oxygen, which we (and others) suggest leads to catalyst modification. Preliminary mechanistic investigation also suggested that the modified catalyst is capable of catalysing the reaction moderately at higher temperatures whilst being ineffective at room temperature. These observations may go some way to explaining why other hydrogen borrowing carbon-carbon bond-forming reactions employ high temperatures and possess limited functional group tolerance. We hope this work will extend the application of the hydrogen borrowing approach in the synthesis of complex and medicinally relevant molecules.

## Methods

### General Procedure A – Hydrogen borrowing reaction with Ph* methyl ketone and [Cp*IrCl$_2$]$_2$

[Cp*IrCl$_2$]$_2$ (2.4 mg, 1 mol%), Ph* methyl ketone (57.1 mg, 0.300 mmol, 1 eq.), an alcohol (if solid, 1 eq.) and powdered potassium *tert*-butoxide (16.8 mg, 0.150 mmol, 0.5 eq.) were added sequentially to a microwave vial equipped with a stir bar. *tert*-Amyl alcohol (0.30 mL or 0.10 mL, 1 M or 3 M respectively) and an alcohol (if liquid, 1.0 eq.) were added quickly and the vial was capped, and evacuated and backfilled with nitrogen (6 × 10 sec). The reaction mixture was then stirred at 23 °C, 85 °C or 115 °C for 18 h. The reaction mixture was filtered through a

short pad of silica gel (elution with diethyl ether, ethyl acetate or methanol depending on assumed relative polarity of the product) and concentrated *in vacuo*. The resulting oil was redissolved in CDCl₃ (2.00 mL) and an NMR standard was added (32 μL 0.30 mmol, 1,1,2,2-tetrachloroethane) to determine yield via ¹H NMR spectroscopy. Further purification was achieved by silica gel column chromatography (elution conditions stated for each compound within the Supplementary Information).

Exemplified for 1-(2,3,4,5,6-Pentamethylphenyl)−3-(3-(pyrrolidin-1-ylmethyl)bicyclo[1.1.1]pentan-1-yl)propan-1-one **23**.

Prepared according to **General Procedure A** with (3-(pyrrolidin-1-ylmethyl)bicyclo[1.1.1]pentan-1-yl)methanol (54.4 mg, 0.300 mmol, 1 eq.) and *tert*-amyl alcohol (0.30 mL, 1 M) at 23 °C. The crude product was purified by silica gel column chromatography (elution with 0–2% methanol in CH₂Cl₂ + 1% triethylamine) to give the title product as a pale-yellow solid (77.2 mg, 73%).

## General Procedure B – Hydrogen borrowing reaction with non-Ph* methyl ketones and [Cp*IrCl₂]₂

[Cp*IrCl₂]₂ (4.8 mg, 2 mol%), a ketone (if solid, 0.300 mmol, 1 eq.), an alcohol (if solid, 0.300 mmol, 1 eq.), powdered potassium *tert*-butoxide (16.8 mg, 0.150 mmol, 0.5 eq.) were added sequentially to a microwave vial equipped with a stir bar. *tert*-Amyl alcohol (3.00 mL, 0.1 M) and an alcohol (if liquid, 0.300 mmol, 1.0 eq.) were added quickly and the vial was capped, evacuated and backfilled with nitrogen (6 × 10 s). The reaction mixture was then stirred at 23 °C, 85 °C or 115 °C for 18 h. The reaction mixture was then filtered through a short pad of silica gel (elution with diethyl ether, ethyl acetate or methanol depending on assumed relative polarity of the product) and concentrated in vacuo. The resulting oil was redissolved in CDCl₃ (2.00 mL) and an NMR standard was added (32 μL 0.30 mmol, 1,1,2,2-tetrachloroethane) to determine yield via ¹H NMR spectroscopy. Further purification was achieved by silica gel column chromatography (elution conditions stated for each compound within the Supplementary Information).

Exemplified for 1-phenyl-3-(3-(pyrrolidin-1-ylmethyl)bicyclo[1.1.1]pentan-1-yl)propan-1-one **60**.

Prepared according to **General Procedure B** with acetophenone (35.0 μL, 0.300 mmol, 1 eq.) and (3-(pyrrolidin-1-ylmethyl)bicyclo[1.1.1]pentan-1-yl)methanol (54.4 mg, 0.300 mmol, 1 eq.) at 23 °C. The crude product was purified by silica gel column chromatography (elution with 0–2% methanol in CH₂Cl₂ + 1% triethylamine) to give the title product as a pale-yellow oil (52.9 mg, 62%).

## General Procedure C – Ph* deprotection with HCl in HFIP

Hydrochloric acid (37% aq. 12 M, 0.13 or 0.26 mL) was added to a solution of substrate (0.100 mmol, 1 eq.) in HFIP (0.88 mL, therefore 2 M or 4 M HCl in HFIP respectively) in a microwave vial equipped with a stir bar. The vial was capped and the reaction solution was stirred at 40 °C or 65 °C for 16 h. The product was then purified with (i) or without (ii) column chromatography as follows:

(i) Chromatography free purification (substrates containing basic nitrogen functionality which is/are protonated under the reaction conditions): the vial was then cooled to rt and water (2 mL) and diethyl ether (3 mL) were added. The aqueous layer was further washed with diethyl ether (4 × 3 mL) and then concentrated in vacuo to give the corresponding carboxylic acid HCl salt.

(ii) Purification by column chromatography (otherwise): H₂O (5 mL) was added, and the reaction mixture was extracted with CHCl₃ (3 × 5 mL); the combined organic extracts were dried using Mg₂SO₄, filtered and concentrated *in vacuo* to yield a crude oil. This oil was purified by silica gel column chromatography (elution conditions stated for each compound within the Supplementary Information).

Exemplified for 1-((3-(2-carboxyethyl)bicyclo[1.1.1]pentan-1-yl)methyl)pyrrolidin-1-ium chloride **87**.

Prepared according to **General Procedure C**, with 3-(1-benzylpiperoxridin-4-yl)−1-(2,3,4,5,6-pentamethylphenyl)propan-1-one (35.4 mg, 0.100 mmol, 1.0 eq.) and hydrochloric acid (37% aq. 12 M, 0.13 mL) at 65 °C. The title product was obtained without column chromatography as a colourless gum (19.2 mg, 74%).

## Data availability

All data (experimental procedures and characterization data) supporting the findings of this study are available within the article and its supplementary information or from the corresponding authors.

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

## Acknowledgements

The EPSRC and Lady Margaret Hall, Oxford have provided financial support for a studentship (to E.P.B.) via the Centre for Doctoral Training in Synthesis for Biology and Medicine (EP/L015838/1). A CC-BY licence is applied to the author accepted manuscript arising from this submission, in accordance with EPSRC's open access conditions. We are grateful to Dr Ben Mckeever-Abbas (AstraZeneca) for helpful discussions.

## Author contributions

E.P.B., T.J.D., and M.D.S. conceived and designed the study; E.P.B. performed the synthetic and mechanistic experiments and analysed data for all compounds. E.P.B., T.J.D., and M.D.S. co-wrote the paper.

## Competing interests

The authors declare no competing interests.
