## [Peer Review File · Nature Communications]

REVIEWERS' COMMENTS

Reviewer #1 (Remarks to the Author):

I provided a favorable evaluation of this manuscript in the initial round of review, and the subsequent improvements made by the authors have made this work even more compelling. Ketone alkylation by alcohols is indeed well-studied, yet often inefficient. The insights provided by the author unlock highly efficient low temperature variants of this important process. Hence, publication of this work is endorsed in the strongest possible terms.

Reviewer #2 (Remarks to the Author):

Donohoe, Smith and Bailey report in this manuscript a C-alkylation of methyl ketones using a practical borrowing hydrogen procedure catalyzed by the commercially available $(Cp^*IrCl)_2$ complex. The highlight of this work is the practicality of the procedure as well as the broad substrate scope bearing various functionalities. The mechanistic studies are rather preliminary and really do not provide in depth understanding of this catalytic system. I agree that this process is a valuable catalytic method to perform α -alkylation of methyl ketones with a wide scope, but I am not convinced this work possesses the novelty for a publication in *Nat. Commun.* In fact the Donohoe group has been recognized as a leader in this field of research and reported valuable catalytic systems to enable C-alkylation of ketones beyond only methyl ketone and also enantioselective variants. In this work, only the discovery of a practical procedure to carry out C-alkylation of methyl ketones really don't justify the significance of this work. If the authors can realize alkylation of a wide range of ketones beyond methyl ketones and even look into enantioselective variant, in my opinion that can be published in *Nat. Commun.*

Reviewer #3 (Remarks to the Author):

In this manuscript, authors described the C-alkylation of ketones with alcohols using hydrogen borrowing reactions. A large number of products containing different substituents were synthesized, which proved the feasibility of the catalytic reaction. The advantage of this reaction is performed in low temperature. Although the original has been revised, reviewer does not think that this manuscript is suitable for the publication in *Nature Communication* due to the following reasons.

1. $[Cp^*IrCl_2]_2$ was first used in hydrogen borrowing reactions (*Tetrahedron Lett.*, 2003, 44, 2687-2690.), a large number of research groups used the same catalyst in the same type of reaction. Thus, the catalytic system lacks novelty.
2. α -alkylation of ketones with alcohols have widely studied. A large amount of transition metal catalysts were applied the reactions, such as *Organic Letters* 2022, 24, 30, 5584-5589; *Organic Letters* 2017, 19, 5, 1080-1083; *ACS Catalysis* 2018, 8, 11, 10300-10305; *Organic Letters* 2019, 21, 19, 8065-8070 ; *Organic Letters* 2018, 20, 18, 5587-5591; *Organic Letters* 2012, 14, 18, 4703-4705. Thus, the type of reactions lacks novelty.

3. Reactions required strong base KOt-Bu (0.5 eq.) and thus a large amount of waster would be formed. This will seriously affect the environment. It does not meet the requirements of green chemistry.
4. This reaction mechanism has not been fully studied.

Response to reviewers' comments (original submission **NCOMMS-24-22911-T**)

We are grateful for the helpful and insightful comments of all three reviewers. We have already replied to some comments from reviewers, but they are repeated below for complete clarity:

Reviewer 1 states *“I provided a favorable evaluation of this manuscript in the initial round of review, and the subsequent improvements made by the authors have made this work even more compelling. Ketone alkylation by alcohols is indeed well-studied, yet often inefficient. The insights provided by the author unlock highly efficient low temperature variants of this important process. Hence, publication of this work is endorsed in the strongest possible terms”*

We are grateful for the positive comments of this reviewer.

Reviewer 2 comments that *“The highlight of this work is the practicality of the procedure as well as the broad substrate scope bearing various functionalities. In this work, only the discovery of a practical procedure to carry out C-alkylation of methyl ketones really don't justify the significance of this work.”*

We recognise that hydrogen borrowing C-alkylation of ketones is a well-investigated transformation and have added a sentence to place our work in a more appropriate context, but respectfully disagree that this is simple. In the literature, functional groups including amines, heterocycles and drug molecules are not described as successful anywhere across the field of hydrogen borrowing. We do not believe that enabling reactions at room temperatures that were previously unsuccessful under any other conditions, and then rationalizing that mechanistically, is “incremental”. For context in real-world applications of hydrogen borrowing see: *Org. Process Res. Dev.* **19**, 1400–1410 (2015). DOI: [10.1021/acs.oprd.5b00199](https://doi.org/10.1021/acs.oprd.5b00199)

Reviewer 2 also states *“The mechanistic studies are rather preliminary and really do not provide in depth understanding of this catalytic system.”*

[see also our response to a related question by reviewer 3]

Our work is focused on the functional group tolerance of hydrogen borrowing C-alkylation – and is not a detailed mechanistic study, which would be outside the scope of this paper. We do directly observe a (precedented) iridium hydride species under the reaction conditions and demonstrate that this species reacts with oxygen; this is to demonstrate consistency with our assertion that oxygen is responsible for catalyst modification and deactivation. As the reviewer notes, we were unable to identify the species produced upon exposure of $[\text{IrCp}^*\text{HCl}]_2$ to oxygen (which they describe as “*nearly impossible*”) but our observations are consistent with the generation of a complex mixture of compounds rather than a single isolable species (Ison and co-workers similarly could not identify products of reaction of $[\text{IrCp}^*\text{HCl}]_2$ with oxygen (*J. Am. Chem. Soc.* **44**, 14462–14464 (2008)). We do not feel that this observation is detrimental – as despite the lack of a clear and identifiable product of the reaction of $[\text{IrCp}^*\text{HCl}]_2$ with oxygen, we are still able to validate the premise that the oxygen sensitivity of the catalytic Ir-H species is key to low temperature reactivity.

Reviewer 3 states that the catalytic system “*lacks novelty*”. $[\text{Cp}^*\text{IrCl}_2]_2$ is a commercially available and air stable precatalyst used in myriad different reactions and so there is no doubt its application in catalysis is not novel (and we obviously do not claim it as such). However, the much-enhanced functional group tolerance described in our paper is not described elsewhere. We screened a range of different metal catalysts (Ir, Ru, Rh, Mn, Fe; see SI pS9) that have been used in hydrogen borrowing alkylation reactions to attain the room temperature reactivity which enables the observed functional group tolerance. The fact that the best catalyst is simple and commercially available is something we view as a benefit (as reviewer 1 notes), especially when such reactivity has not been reported before.

The same reviewer then goes on to say “*The alpha-alkylation of ketones with alcohols have widely studied. A large amount of transition metal catalysts were applied the reactions, such as Organic Letters 2022, 24, 30, 5584-5589 ; Organic Letters 2017, 19, 5, 1080-1083; ACS Catalysis 2018, 8, 11, 10300-10305; Organic Letters 2019, 21, 19, 8065-8070 ; Organic Letters 2018, 20, 18, 5587-5591; Organic Letters 2012, 14, 18, 4703-4705. Thus, the type of reactions lacks novelty.*”

We agree with this reviewer that the α -alkylation of ketones with alcohols is a transformation that has been intensively investigated. As we state in the manuscript: “*This process is exemplified by the alkylation of acetophenone with benzyl alcohol, which has been achieved with a wide range of different metal catalysts*”, before citing two comprehensive reviews. In figure 1, we also list 10 metals that have been used as components of catalysts to mediate this specific reaction. We do recognise that could be more explicit in stating this and hence have added greater context for our work by adding a sentence stating “*the α -alkylation of ketones via hydrogen borrowing has been intensively investigated*” and citing all the papers suggested by this reviewer (now references 6-10 and 15 in the manuscript). It is also worth noting that functional group tolerance is uncommon across the field of hydrogen borrowing, and these papers broadly exemplify this; references 6-10 generally employ temperatures of 110-140 °C for alcohol substrates devoid of *any* sensitive functional groups.

Reviewer 3 goes on to state: “*reactions required strong base KOt-Bu (0.5 eq.) and thus a large amount of waster would be formed. This will seriously affect the environment. It does not meet the requirements of green chemistry*”.

We agree with this reviewer that the use of lower quantities of reagents would be beneficial to the environment. However, we use substoichiometric base, and for context, only two of the six studies cited by this reviewer use less base (%) than our study. None of them outline the environmental impact of any method, and so we feel this is not a viable metric by which to judge this manuscript.

This reviewer goes on to state “*This reaction mechanism has not been fully studied*”.

We agree that the reaction mechanism has not been fully studied – but this is not the focus of this paper. For context, to our knowledge there are no detailed published studies of the mechanism of *any* synthetic hydrogen borrowing reaction. In fact, we cannot find *any* hydrogen borrowing study that explicitly describes the observation of an iridium hydride intermediate of *any* sort. What we have done is propose a mechanistic basis why our system is more functional group tolerant than others and provided some experimental evidence to support it; there is doubtless more work to do here but we feel much of this lies outside the context of this current synthetic study.